# Does Post-Activation Performance Enhancement Occur during the Bench Press Exercise under Blood Flow Restriction?

**DOI:** 10.3390/ijerph17113752

**Published:** 2020-05-26

**Authors:** Michal Wilk, Michal Krzysztofik, Aleksandra Filip, Agnieszka Szkudlarek, Robert G. Lockie, Adam Zajac

**Affiliations:** 1Institute of Sport Sciences, The Jerzy Kukuczka Academy of Physical Education in Katowice, 40-065 Katowice, Poland; m.krzysztofik@awf.katowice.pl (M.K.); a.filip@awf.katowice.pl (A.F.); a.zajac@awf.katowice.pl (A.Z.); 2Department of Physical Pharmacy, Faculty of Pharmaceutical Sciences in Sosnowiec, Medical University of Silesia in Katowice, 40-055 Katowice, Poland; aszkudlarek@sum.edu.pl; 3Center for Sports Performance, Department of Kinesiology, California State University, Fullerton, CA 92831, USA; rlockie@fullerton.edu

**Keywords:** upper limb resistance exercise, power output, bar velocity, occlusion, performance

## Abstract

*Background*: The aim of the present study was to evaluate the effects of post-activation performance enhancement (PAPE) during successive sets of the bench press (BP) exercise under blood flow restriction (BFR). *Methods*: The study included 10 strength-trained males (age = 29.8 ± 4.6 years; body mass = 94.3 ± 3.6 kg; BP 1-repetition maximum (1RM) = 168.5 ± 26.4 kg). The experiment was performed following a randomized crossover design, where each participant performed two different exercise protocols: under blood flow restriction (BFR) and control test protocol (CONT) without blood flow restriction. During the experimental sessions, the study participants performed 3 sets of 3 repetitions of the BP exercise at 70%1RM with a 5 min rest interval between sets. The differences in peak power output (PP), mean power output (MP), peak bar velocity (PV), and mean bar velocity (MV) between the CONT and BFR conditions were examined using 2-way (condition × set) repeated measures ANOVA. Furthermore, t-test comparisons between conditions were made for the set 2–set 1, set 3–set 1, and set 3–set 2 delta values for all variables. *Results*: The post hoc results for condition × set interaction in PP showed a significant increase in set 2 compared to set 1 for BFR (*p* < 0.01) and CONT (*p* = 0.01) conditions, a significant increase in set 3 compared to set 1 for the CONT (*p* = 0.01) condition, as well as a significant decrease in set 3 compared to set 1 for BFR condition occurred (*p* < 0.01). The post hoc results for condition × set interaction in PV showed a significant increase in set 2 compared to set 1 for BFR (*p* < 0.01) and CONT (*p* = 0.01) conditions, a significant increase in set 3 compared to set 1 for CONT (*p* = 0.03) condition, as well as a significant decrease in set 3 compared to set 1 for BFR condition (*p* < 0.01). The t-test comparisons showed significant differences in PP (*p* < 0.01) and PV (*p* = 0.01) for set 3–set 2 delta values between BFR and CONT conditions. *Conclusion*: The PAPE effect was analyzed through changes in power output and bar velocity that occurred under both the CONT and BFR conditions. However, the effects of PAPE have different kinetics in successive sets for BFR and for CONT conditions.

## 1. Introduction

The ability to generate high power outputs is one of the most significant factors determining performance in numerous sport disciplines. The increase in power output observed during and directly after resistance exercises is termed post-activation performance enhancement (PAPE) [1]. Previous studies have shown a ergogenic effect of initial muscle activation on power output in consecutive sets of a single resistance exercise [2,3] in addition to a stimulating effect of initial muscle activation on effectiveness of throwing, jumping, and hitting movements as well on the time under tension during exercise performed to volitional failure [4,5,6,7]. The physiological rationale for PAPE is a greater contractibility and excitability of muscles due to several mechanisms [2,8]. The rationale for this phenomenon might be related to residual of post-activation potentiation (PAP) in its earliest stages after a conditioning activity (CA) and other mechanisms such as enhanced muscle activation, increased muscle temperature and muscle force, and/or increased shortening velocity triggered by fiber water content [8]. 

Most studies that have analyzed the acute effects of PAPE on power output have considered training components such as external load, number of sets and repetitions, as well as the duration of rest intervals between particular sets and between the conditioning and explosive exercise [5,9,10,11]. Regarding the intensity a wide range of loads have been found to potentiate subsequent performance (40–90% 1-repetition maximum (1RM)) [12,13,14,15]. In addition to the applied external load, the number of sets during the conditioning activity also have a significant impact on PAPE [3,16]. One of the newly introduced training means in resistance exercise includes occlusion, which is better known as blood flow restriction (BFR) [17]. The BFR technique involves the use of a tourniquet, inflatable cuff, or elastic wraps [18,19]. The compression is placed at the upper part of the limb to reduce arterial blood flow and to occlude venous blood flow during physical exercise [17]. The main mechanisms responsible for the adaptive responses related with exercise under BFR include increased mechanical tension and elevated metabolic stress [20]. Although the exact mechanisms remain unknown, most evidence seems to allude to a muscle cell swelling response and the indirect effect of metabolites, instigating an increased muscle activation through fatigue [21]. Further, the exercise under BFR resulted in greater muscle cross-sectional area, isometric strength, and 1RM performance [22]. Application of BFR acutely increases muscle swelling due to fluid shifts [18]. Given that myofibrillar fluid shifts can increase muscle fiber force and shortening velocity and, thus, power, this mechanism can increase effectiveness of PAPE [8]. Furthermore, Wilk et al. [23] suggest that not only the physiological responses but also mechanical work generated by the cuff can potentially cause acute positive changes during resistance exercise under BFR. Currently only few previous studies have compared the effects of high load resistance training under BFR conditions [23,24,25,26], and only one refers to acute changes in power output [23]. The study by Wilk et al. [23] showed that short-term BFR increases power output and bar velocity during the bench press (BP) exercise. 

While PAPE alone offers an attractive and practical means for conditioning coaches to elicit an increase in performance, the additional use of BFR during resistance training can additionally enhance the PAPE effect. Considering this, numerous studies have attempted to determine optimal variables of resistance training that would maximize PAPE [3,27,28]. However, none of them relate to the acute changes in power output following successive sets of resistance exercise under BFR. Due to the fact that short-term BFR significantly increases the power output and bar velocity [23], the aim of the present study was to evaluate the effects of BFR on power output and bar velocity between successive sets of the BP exercise. We hypothesized that the PAPE effect occurs during successive sets of the BP exercise under BFR.

## 2. Methods

The experiment was performed following a randomized crossover design, where each participant performed two different exercise protocols: under BFR with a 10 cm cuff (BFR) and a control test protocol (CONT) without BFR. The entire research procedure lasted 4 weeks with a one-week interval between each trial. During the experimental sessions, the study participants performed 3 sets of the BP exercise of 3 repetitions at 70% of 1RM with a 5 min rest interval between sets at maximum speed. The following variables were registered: peak power output (PP), mean power output (MP), peak bar velocity (PV), and mean bar velocity (MV). All testing sessions were performed in the Strength and Power Laboratory at the Academy of Physical Education in Katowice.

### 2.1. Participants

Ten healthy males who were experienced in resistance training (12.7 ± 6.8 years) volunteered for the study (age = 29.8 ± 4.6 years; body mass = 94.3 ± 13.6 kg; BP 1RM = 168.5 ± 26.4 kg; mean ± SD). The inclusion criterion was a BP personal record with a load of at least 120% body mass. The participants were instructed to maintain their normal dietary habits over the course of the study and not to use any supplements or stimulants for the duration of the experiment. All participants were required to refrain from resistance training 72 h prior to each experimental session, and were informed about the benefits and potential risks of the study before providing their written informed consent for participation. The participants could withdraw from the experiment at any moment of the study. The study protocol was approved by the Bioethics Committee for Scientific Research at the Academy of Physical Education in Katowice, Poland (no. 02/2019), and performed according to the ethical standards of the latest version of the Declaration of Helsinki, 2013. 

### 2.2. Familiarization Session and One Repetition Maximum Test

Two weeks before the main experiment, the participants took part in familiarization sessions consisting of 4 sets of 3 repetitions of the BP at 50%1RM with BFR with cuff pressure set to the value of ~60% full arterial occlusion pressure (AOP).

One week before the main experiment, all participants performed the 1RM BP test. The general warm-up before the 1RM test consisted of cycling on an ergometer for 5 min, followed by several push-ups, pull-ups, and dynamic stretching of the upper body. Next, the participants performed 15, 10, and 5 BP repetitions using 20%, 40%, and 60% of their estimated 1RM, respectively. For the evaluation of 1RM, the loading started at 80% estimated 1RM, and was increased by 2.5 to 10 kg for each subsequent attempt, and the process was repeated until failure. During the 1RM test, the participants executed single repetitions with a constant movement tempo (2 s duration of the eccentric phase, maximal speed in the concentric phase) [29,30] and a 5 min rest interval between successful attempts. Hand placement on the bar was set at 150% bi-acromial distance. 

### 2.3. Experimental Sessions

In a randomized, crossover fashion, the participants performed the BP exercise at 70%1RM either with BFR (BFR) or without BFR (CONT). The general and specific warm-up for the experimental sessions was identical to the one used during the 1RM test. After the warm-up, the participants started the main protocol and performed 3 sets of 3 repetitions of concentric and eccentric contractions at maximal tempo of movement with a 5 min rest interval between sets. The repetitions were performed without intentionally pausing at the transition between the eccentric and concentric phases. A linear position transducer system (Tendo Sport Machines, Trencin, Slovakia) was used for the evaluation of bar velocity. The Tendo Power Analyzer is a reliable system for measuring movement velocity and power output (commercially calibrated) [31]. The measurement was made independently for each repetition and automatically converted into values of peak power output (PP), mean power output (MP), peak velocity (PV), and mean velocity (MV). The mean power output and bar velocity were obtained as the mean of the three repetitions. Peak power output and peak bar velocity were obtained from the best repetition. 

### 2.4. Blood Flow Restriction 

The participants wore occlusion cuffs at the most proximal region of both arms during experimental sessions. During the exercise protocol with BFR, Smart Cuffs (10 cm width) produced by Smart Tools Plus LLC, Strongsville, USA were applied to the upper limbs. In order to determine the individual occlusion pressure, the value of full AOP at rest was determined. The measurement was conducted twice on each limb, and the obtained differences were within 20 mmHg [32]. The average value was then used to set the cuff pressure for the exercise protocol. The pressure of the cuff for BFR condition was set to ~90% of full AOP (152 mmHg ± 11.4 for BFR) [23]. The level of vascular restriction was controlled by a handheld Edan SD3 Doppler (Edan Instruments, Shenzen, China). During the BFR condition, the restriction was set immediately before the onset of exercise and released following the completion of the third repetition [23]. 

### 2.5. Statistical Analysis

All statistical analyses were performed using Statistica 9.1 (Hillview, Palo Alto, CA, USA) and were presented as means with standard deviations. The Shapiro–Wilk and Mauchly’s tests were used in order to verify the normality/homogeneity and sphericity of the sample data variances, respectively. Verification of differences between CONT and BFR conditions in PP, MP, PV, and MV was performed using a two-way 2 × 3 (condition × set) analysis of variance (ANOVA) with repeated measures. Statistical significance was set at *p* < 0.05. In the event of a significant main effect, post hoc comparisons were conducted using Tukey’s test. Furthermore, t-test comparisons between conditions were made for the set 2–set 1, set 3–set 1, and set 3–set 2 delta values for all variables. Additionally, independent sample t-tests were used to verify the differences between successive sets independently for BFR and CONT conditions as well as differences in individual sets between BFR and CONT conditions. Percent changes and 95% confidence intervals were also calculated. Effect sizes (Cohen’s *d*) were reported where appropriate. Parametric effect sizes (ES) were defined as large *d* > 0.8; moderate between 0.79 and 0.5; small between 0.49 and 0.20; and trivial as <0.2 [33].

## 3. Results 

The two-way repeated measures ANOVA indicated significant condition × set interaction effect for PP (*p* = 0.01) and PV (*p* = 0.03). There was no significant condition × set interaction effect for MP (*p* = 0.08) and MV (*p* = 0.11). Furthermore, there was a significant main effect of condition in PP, MP, PV and MV.

The post hoc results for condition × set interaction in PP showed a significant increase in set 2 compared to set 1 for BFR (*p* < 0.01) and CONT (*p* = 0.01) conditions, a significant increase in set 3 compared to set 1 for the CONT (*p* = 0.01) condition, as well as a significant decrease in set 3 compared to set 1 for BFR condition (*p* < 0.01). The post hoc results for the condition × set interaction in PV showed a significant increase in set 2 compared to set 1 for BFR (*p* < 0.01) and CONT (*p* = 0.01) conditions, a significant increase in set 3 compared to set 1 for CONT (*p* = 0.03) condition, as well as a significant decrease in set 3 compared to set 1 for BFR condition (*p* < 0.01). Further, the post hoc results for the condition × set interaction in PP and PV showed a significant increase in set 1, set 2, and set 3 for BFR compared to CONT condition *p* < 0.01 for all; Table 1; Figure 1, Figure 2, Figure 3 and Figure 4).

The post hoc results for the main effect of particular conditions showed a significant increase in PP (792 vs. 965 W), MP (559 vs. 667 W), PV (0.62 vs. 0.74 m/s), and MV (0.46 vs. 0.53 m/s) for BFR when compared to the CONT (*p* < 0.01 for all).

The t-test comparisons for delta values showed significant differences in PP (*p* < 0.01) and PV (*p* = 0.01) for set 3–set 2 between BFR and CONT conditions (Table 2).

The results of the t-test used to compare differences between sets and between conditions are presented in Table 3 and Table 4. 

## 4. Discussion 

The main finding of the study was that the PAPE effect can occur during the BP exercise under BFR. The results showed that the PAPE effect analyzed by changes in PP and PV occurred under both the CONT as well as BFR conditions. However, based on the direct analysis of changes between particular sets, significant differences were observed in result of set 3–set 2 between CONT and BFR conditions. Furthermore, the results of the present study show significant increases in power output and bar velocity during the BP exercise for the BFR condition compared to the CONT condition, which confirms previous results [23].

The main goal of this study was to evaluate the changes in power output and bar velocity between successive sets of the BP exercise, with and without BFR. This is the first study to address the effect of PAPE during high load resistance exercise under BFR which limits the possibility of comparing our results to other studies. Previous research has shown that resistance exercise under BFR increases metabolic stress compared to traditional resistance training [34]. However, most studies with BFR relate to chronic adaptive changes [34,35,36]. Only the study by Wilk et al. [23] analyzed acute changes in power output and bar velocity between exercises performed without and under BFR. The study of Wilk et al. [23] showed that short-term and high-pressure BFR increased power output and bar velocity during the BP compared to exercise without BFR. To date, two studies [37,38] examined the use of BFR on PAPE. In the study by Cleary and Cook [38], the use of BFR (60%AOP) led to diminished vertical jump performance after a CA at 30%1RM, while Miller et al. [37] showed increased vertical jump height after a 10 s maximal isometric deadlift combined with BFR. However, in both studies, BFR was used only during the CA and was removed for the post-activation exercise. Previous studies showed that the main adaptive changes during exercise with BFR are related to increased metabolic stress [35,36]. The increased metabolic stress following exercise with BFR results from the accumulation of metabolic products of physical activity in the part of the limb that is restricted from blood flow [24,39]. However, the increase in metabolic stress following exercise with BFR probably does not refer to short-term occlusion [23], as has been conducted in the present study. The occlusion applied in our experiment lasted only a few seconds (3 repetitions; maximal tempo of movement; ~5 s) during each set of the BP exercise. The short duration of the effort was dictated by the predominance of anaerobic metabolism; therefore, the metabolic stress associated with training under BFR conditions was probably not very intense. 

Interestingly, a detailed analysis of the differences between particular sets indicated that the PAPE effect has different kinetics between particular sets for the BFR and CONT conditions. Significantly different kinetics for BFR and CONT conditions were observed in PP and PV values between set 3 and set 2. For the BFR condition, PP and PV were increased in set 2 compared to set 1, and decreased in set 3 compared to set 2. However, the decrease was not observed in the CONT condition. It is possible that benefits from a more pronounced PAPE in set 2 in the BFR condition could be counterbalanced by more pronounced muscle exhaustion in set 3. The short-term BFR increased the absolute value of power output and bar velocity during the BP exercise [23], however, such an increase of performance may promote greater fatigue which, as a consequence, may cause a decrease in power and related variables in subsequent sets of the exercise, as observed in the present study. However, despite the observed significant decrease in PP and PV in set 3–set 2 for the BFR condition, the absolute value of power output and bar velocity was still significantly higher when compared to the CONT condition, which suggests the BFR is an effective tool in acute enhancement of power output and bar velocity during several sets of the BP exercise. 

These results indicated that short-term, high-pressure BFR has a significant impact on the level of acute changes during resistance exercise. The results of our study are partially consistent with findings presented by Morales-Artacho et al. [16] and Wilk et al. [3], who documented increases in power output during a resistance exercise in the second set compared to the first. However, contrary to Morales-Artacho et al. [16] and Wilk et al. [3], we also observed significant increases in power output and bar velocity in the third set compared to first and, thus, the BFR condition did not show such an significant advantage. The ratio of the duration of the effort during the rest interval is an important factor in PAPE exercise protocols [3,9,40]. An optimal rest interval should ensure the optimal balance between fatigue and potentiation [3,9]. Wilk et al. [3] showed that longer duration of repetitions limit PAPE effectiveness in successive sets due to insufficient rest and accumulated fatigue. Therefore, it can also be assumed that during the BFR condition, the lack of PAPE benefit in set 3–set 2 may be associated with an insufficient rest interval and fatigue. The higher absolute values of power output and bar velocity during the BFR condition as well as the fact that blood flow restriction may potentially promote greater fatigue may indicate the need for a longer rest interval between activation and potentiation [3,9]. Nevertheless, there are no data and guidelines for optimizing the rest interval for increased power performance during exercise under BFR, which requires further research. 

To induce the PAPE effect, it is necessary to use high loads [9,10,11,12]. A combination of high loading (70%1RM) and BFR with high pressure (90%AOP) is not compatible with the recommendations of Patterson et al. [32]. Furthermore, Counts et al. [41] showed that higher relative pressures (90%AOP) may not be necessary when exercising under BFR. However, it should be noted that during the present study, we used intermittent and not continuous occlusion. Furthermore, the occlusion applied lasted only a few seconds during a set (~15 s for the whole training session), while the recommendations by Patterson et al. [32] for resistance exercise under BFR assume a restriction time of 5–10 min with reperfusion between exercises, not between sets. Currently, there are no guidelines for determining the optimal pressure as well as the time of restriction for exercise under BFR when the goal is acute increases of strength or power performance. Furthermore, a recent study by Wilk et al. [42] has shown the positive effect of extremely high pressure (150%AOP) on acute strength and strength endurance performance enhancement. This study suggested that not only BFR but also external muscle compression and mechanical energy generated by the cuff may be an important factor in the effectiveness of occlusion during the resistance exercise [23].

The type of exercise chosen for the study protocol may be a factor determining the level of acute changes during resistance exercise under BFR. In the present study, muscle occlusion was used at the upper limb (arms), while the main muscles involved in the BP are the pectoralis major and anterior deltoid [43]. The triceps brachii muscle shows high activity during BP [43], but BFR in this area does not affect the changes taking place in the pectoralis major and deltoid muscles, the two other primary muscles involved in the BP exercise. Despite the fact that Yasuda et al. [44] suggested that BP training under BFR also leads to an increase in muscle size in the area of the chest muscles, the presented results were related to acute and not chronic responses. Therefore, it can be assumed that the use of a long-term training program or another exercise, especially an exercise involving only those muscles that are occluded, could cause different, even conflicting results to those presented in this study. 

The present study has some limitations which should be addressed. Although the results showed that the PAPE effect occurred during resistance exercise under BFR, the direct causes of these changes cannot be determined and explained. There was no analysis of direct physiological changes that would be the basis for explaining the obtained results. Furthermore, it is possible that the BFR increases local blood volume and therefore increases skin temperature [45], which may be related to performance enhancement [46,47]; however, skin temperature was not recorded in our study. Finally, the results of our study refer only to PAPE effects of the upper limbs during the BP exercise, and only to the cuff pressure used, and cannot be translated into other exercises, volumes, or intensities nor other occlusion pressures. Therefore, further research is required, especially in assessing the acute impact of continuous and intermittent BFR on the PAPE, during different training protocols as well as with different cuff pressures. 

### Practical Implications

The short-term BFR increases power output and bar velocity during the BP exercise compared to CONT conditions. Such an acute increase under BFR conditions compared to the CONT occurred in all three sets of the BP exercise. However, there are differences in the kinetics of power output and bar velocity changes between BFR and CONT conditions especially in the difference between set 3 and set 2. Under BFR, we observed an increase in the second set compared to the first, but in the third set, the power decreased compared to the second set. On the contrary, in the CONT condition, power output and bar velocity were maintained in the third set compared to the first. Therefore, given a real exercise program, the use of BFR and its effect on the increase in acute power performance should be controlled by measuring devices in order to individually adjust the optimal number of sets and time interval to the induced PAPE. Furthermore, even if an increase in power output and bar velocity occurs as a result of BFR, this effect may not necessarily translate into long-term effects or may even hinder performance. The frequent use of BFR can impair the muscle structure directly in the region under the cuff, which can decrease sport performance and increase the risk of injury [48,49]. Therefore, resistance exercise under BFR should be used occasionally and be well periodized.

## 5. Conclusions

This study demonstrated that the PAPE effect occurs during successive sets of the BP exercise under BFR at 70%1RM. However, compared to the CONT condition, we observed different kinetics of power output and bar velocity changes during successive sets of the BP exercise. Therefore, the application BFR during resistance training can introduce, a new, additional tool in the development of power output, which opens opportunities for modification of strength training programs, particularly in elite athletes. 

## Figures and Tables

**Figure 1 ijerph-17-03752-f001:**
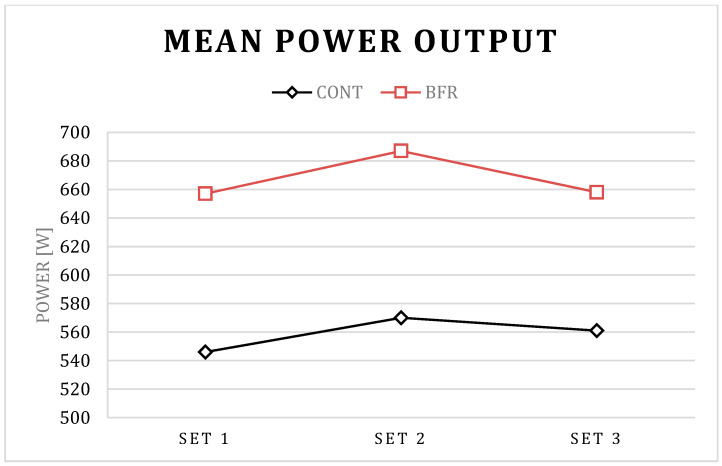
Mean power output for the BFR and CONT conditions during three successive sets of the bench press exercise.

**Figure 2 ijerph-17-03752-f002:**
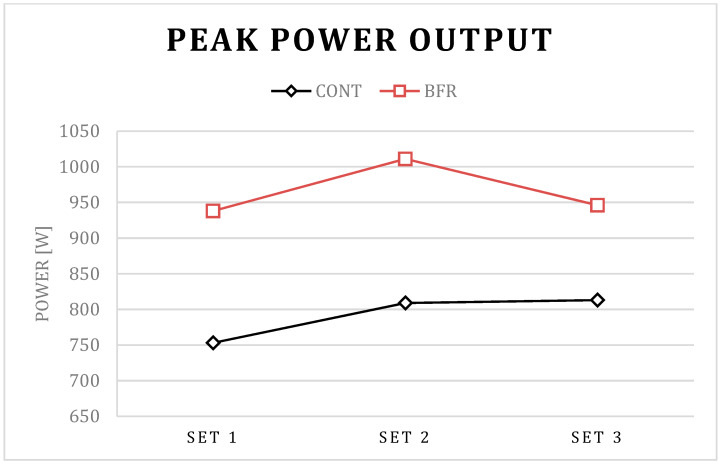
Peak power output for the BFR and CONT conditions during three successive sets of the bench press exercise.

**Figure 3 ijerph-17-03752-f003:**
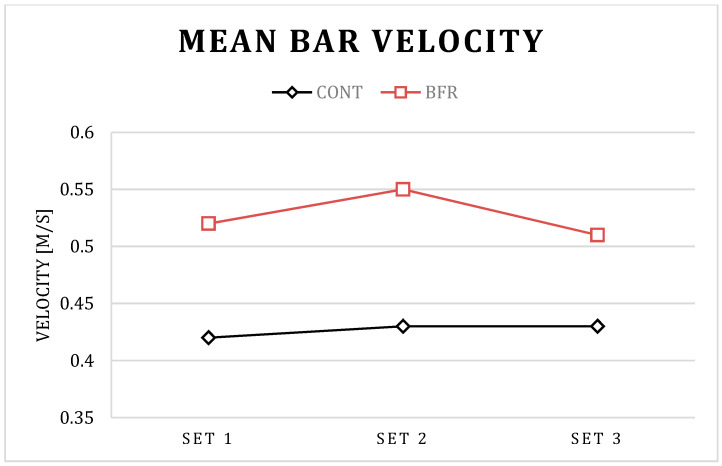
Mean bar velocity for BFR and CONT conditions during three successive sets of the bench press exercise.

**Figure 4 ijerph-17-03752-f004:**
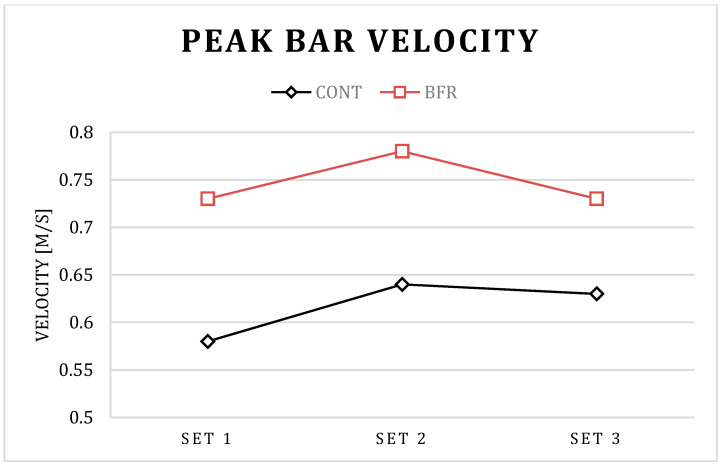
Mean bar velocity for BFR and CONT conditions during three successive sets of the bench press exercise.

**Table 1 ijerph-17-03752-t001:** Power output and bar velocity in 3 successive sets of the bench press exercise under CONT and BFR conditions.

	Set 1	Set 2	Set 3
Peak Power Output (W)			
CONT (95% CI)	753 ± 6 (708 to 799)	809 ± 100 (738 to 880)	813 ± 93 (746 to 879)
BFR (95% CI)	938 ± 82 (879 to 996)	1011 ± 103 (938 to 1085)	946 ± 105 (871 to 1021)
Mean Power Output (W)			
CONT (95% CI)	546 ± 55( 507 to 585)	570 ± 6 (522 to 617)	561 ± 79 (505 to 617)
BFR(95% CI)	657 ± 92 (591 to 723)	687 ± 105 (611 to 762)	658 ± 86 (596 to 719)
Peak Bar Velocity (m/s)			
CONT (95% CI)	0.58 ± 0.09 (0.52 to 0.64)	0.64 ± 0.10 (0.56 to 0.71)	0.63 ± 0.09 (0.56 to 0.69)
BFR (95% CI)	0.73 ± 0.06 (0.68 to 0.77)	0.78 ± 0.08 (0.72 to 0.83)	0.73 ± 0.10 (0.65 to 0.80)
Mean Bar Velocity (m/s)			
CONT (95% CI)	0.42 ± 0.08 (0.36 to 0.48)	0.43 ± 0.07 (0.38 to 0.48)	0.43 ± 0.06 (0.38 to 0.47)
BFR (95% CI)	0.52 ± 0.07 (0.47 to 0.57)	0.55 ± 0.08 (0.49 to 0.60)	0.51 ± 0.08 (0.46 to 0.57)

CONT—control condition; BFR—blood flow restriction condition.

**Table 2 ijerph-17-03752-t002:** The comparison of differences in particular sets between BFR and CONT conditions.

Bench Press	CONT	BFR	Mean Difference	95% CI for Difference	*p*	ES
Differences between sets	Mean Power Output					
Set 2–set 1	23.7 ± 28.2	29.1 ± 16.5	5.4	−27.43 to 16.63	0.59	0.23
Set 3–set 1	15.1 ± 48.4	0.5 ± 27.3	14.6	−34.5 to 63.6	0.52	0.37
Set 3–set 2	−8.6 ± 24.8	−28.6 ± 34.8	20.0	−12.6 to 52.6	0.20	0.66
Differences between sets	Peak Power Output					
Set 2–set 1	55.7 ± 40.7	73.7 ± 56.8	18.0	−54.5 to 18.5	0.29	0.36
Set 3–set 1	59.2 ± 44.2	8.7 ± 71.1	50.5	−10.1 to 111.1	0.09	0.85
Set 3–set 2	3.5 ± 29.9	−65.0 ± 39.4	68.5	24.8 to 112.2	0.01*	1.96
Differences between sets	Mean Bar Velocity					
Set 2–set 1	0.015 ± 0.025	0.028 ± 0.019	0.013	−0.034 to 0.008	0.20	0.59
Set 3–set 1	0.007 ± 0.058	−0.007 ± 0.020	0.014	−0.039 to 0.067	0.57	0.32
Set 3–set 2	−0.008 ± 0.040	−0.035 ± 0.033	0.027	−0.014 to 0.068	0.17	1.15
Differences between sets	Peak Bar Velocity					
Set 2–set 1	0.054 ± 0.033	0.053 ± 0.029	0.001	−0.029 to 0.031	0.94	0.03
Set 3–set 1	0.042 ± 0.037	−0.001 ± 0.053	0.043	−0.004 to 0.089	0.07	0.94
Set 3–set 2	−0.012 ± 0.017	−0.054 ± 0.037	0.042	0.006 to 0.078	0.03 *	1.46

Mean ± standard deviation (SD); * statistically significant difference *p* < 0.05; CONT—control condition; BFR—blood flow restriction condition.

**Table 3 ijerph-17-03752-t003:** A comparison between particular sets of the bench press exercise based on t-test results.

	Peak Power Output			Mean Power Output			Peak Velocity			Mean Velocity		
Bench Press	Mean Difference	*p*	Effect Size	Mean Difference	*p*	Effect Size	Mean Difference	*p*	Effect Size	Mean Difference	*p*	Effect Size
**CONT**												
Set 2 vs. Set 1	74	0.01 *	0.67	24	0.03 *	0.40	0.06	0.01 *	0.63	0.01	0.09	0.13
Set 3 vs. Set 1	78	0.01 *	0.76	15	0.35	0.22	0.05	0.01 *	0.55	0.01	0.71	0.14
Set 3 vs. Set 2	4	0.72	0.04	9	0.30	0.12	0.01	0.06	0.10	0.00	0.54	0.00
**BFR**												
Set 2 vs. Set 1	73	0.01 *	0.78	30	0.01 *	0.30	0.05	0.01 *	0.71	0.03	0.01 *	0.40
Set 3 vs. Set 1	8	0.71	0.08	1	0.95	0.01	0.00	0.95	0.00	0.01	0.30	0.13
Set 3 vs. Set 2	65	0.01 *	0.62	29	0.03 *	0.30	0.05	0.01 *	0.55	0.04	0.01 *	0.50

Mean ± standard deviation (SD); * statistically significant difference *p* < 0.05; CONT—control condition; BFR—blood flow restriction condition.

**Table 4 ijerph-17-03752-t004:** A comparison in individual sets between conditions based on t-test results.

	Peak Power Output			Mean Power Output			Peak Velocity			Mean Velocity		
	Mean Difference	*p*	Effect Size	Mean Difference	*p*	Effect Size	Mean Difference	*p*	Effect Size	Mean Difference	*p*	Effect Size
**CONT vs. BFR**												
Set 1	185	0.01 *	3.18	111	0.01 *	1.46	0.15	0.01 *	1.96	0.10	0.01 *	1.33
Set 2	202	0.01 *	1.99	117	0.01 *	1.57	0.14	0.01 *	1.55	0.12	0.01 *	1.60
Set 3	133	0.01 *	1.34	97	0.01 *	1.17	0.10	0.01 *	1.05	0.08	0.01 *	1.13

Mean ± standard deviation (SD); * statistically significant difference *p* < 0.05; CONT—control condition; BFR—blood flow restriction condition.

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
