# Peer review of "Does Post-Activation Performance Enhancement Occur during the Bench Press Exercise under Blood Flow Restriction?"

_ijerph, 2020, doi:10.3390/ijerph17113752_

Round 1

Reviewer 1 Report

Congratulations to the authors for coming up with research with a recent term on Post Activation Performance Enhancement – PAPE. It was a challenge and a pleasure to review this manuscript.

Since it has become known only a few years ago by Cuenca- Fernandez and colleagues, authors should refer to its acronym in the title, as well as, if possible, to include the definition in the introduction. It is not uncommon to both terms to be confounded despite literature and research such as this one contributes to unbreak the differences. In this way, in line 42, after post-activation potentiation, please add (PAP).

At line 71, change “resentence” to “resistance…”

At line 93, please note, “The participants had the right to withdraw…”

Since several variables affect the PAPE, namely the warm-up (full or comprehensive), it should be possible for readers to know if there was a warm-up before the experimental sessions. It looks like that no warm-up was made, but the information is lacking for the reader’s further comprehension. It there was a warm-up, please describe it.

Regarding the BFR protocol, two values called attention, 90% of AOP with the 70% 1RM load. Generally, with low loads, higher AOP is needed to elicit training effects. In this case, the authors used both high loads and AOP. It is known that PAPE needs high-intensity voluntary conditioning contractions to enhance performance. On the other hand, the recent paper referring to methods, application, and safety (see Patterson et al., 2019) points to the use of BFR with more conservative values of AOP and %1RM. If possible, include some explanation in the discussion regarding this topic.

At line 142 please add a coma after 0.5

At line 145, please note “…for BFR conditions, as well as statistically significant differences in PP and PV…”

Regarding the results, they are well presented. Consider a more visual output (graphic or figure). It should point out the results more rapidly

In the discussion, authors considered the PAPE effect more pronounced during upper limb resistance exercise under BFR only because of the lack of differences in MP and MV in the CONT group. Would not the absolute differences (seems that were not tested for differences) between the conditions be sufficient to point out that more pronounced effect? As in all sets, in all variables, the BFR protocol values are always higher than control.

At line 201 erase “to” in “during to the rest”

At line 205, for further information check Patterson et al. (2019).

The limitations are very descriptive and specified, helping on future research. Please pay attention to the state authors make at line 179 “The metabolic stress associated with training under BFR conditions, was not achieved in the current study, due to the short duration of effort in our exercise protocol”, added by the 5 min inter-set rest; and then at line 216 “Therefore, the use of BFR in upper limbs cannot affect the metabolic stress in the pectoralis major and deltoid muscles.” If the short duration and long inter-set rest precluded the metabolic stress of the blood flow restricted muscles, it might do the same to the pectoralis major and deltoid muscles.

An interesting part of this study that will raise many doubts for further investigations is the fact that BFR did not allow its biggest argument, which is metabolic stress since the restriction was applied for a short time and especially because the intervals between sets were long. This doubt will be kept and should be explored in the future.

Would the authors consider not measuring muscle temperature a limitation?

A section of practical applications will be welcome for readers and practitioners.

Author Response

Congratulations to the authors for coming up with research with a recent term on Post Activation Performance Enhancement – PAPE. It was a challenge and a pleasure to review this manuscript.

Thank you for your thoughtful and careful review of the manuscript. We believe that your suggestions and our revision will improve the quality of the manuscript. You will find your comments below followed by our point-by-point responses.

Since it has become known only a few years ago by Cuenca- Fernandez and colleagues, authors should refer to its acronym in the title, as well as, if possible, to include the definition in the introduction. It is not uncommon to both terms to be confounded despite literature and research such as this one contributes to unbreak the differences. In this way, in line 42, after post-activation potentiation, please add (PAP).

Reply – change has been made. (L – 46)

At line 71, change “resentence” to “resistance…”

Reply - the change has been made.

At line 93, please note, “The participants had the right to withdraw…”

Reply - the change has been made.

Since several variables affect the PAPE, namely the warm-up (full or comprehensive), it should be possible for readers to know if there was a warm-up before the experimental sessions. It looks like that no warm-up was made, but the information is lacking for the reader’s further comprehension. It there was a warm-up, please describe it.

Reply – the information about the warm-up protocol before the experimental session was added. (L 115-117)

Regarding the BFR protocol, two values called attention, 90% of AOP with the 70% 1RM load. Generally, with low loads, higher AOP is needed to elicit training effects. In this case, the authors used both high loads and AOP. It is known that PAPE needs high-intensity voluntary conditioning contractions to enhance performance. On the other hand, the recent paper referring to methods, application, and safety (see Patterson et al., 2019) points to the use of BFR with more conservative values of AOP and %1RM. If possible, include some explanation in the discussion regarding this topic.

Reply – as suggested, a discussion of our experimental procedure regarding the Patterson et al., 2019 recommendation has been added. (L 248-261)

At line 142 please add a coma after 0.5

Reply – a coma was added.

At line 145, please note “…for BFR conditions, as well as statistically significant differences in PP and PV…”

Reply – according to the suggestion of the editor and reviewer the statistical analysis was completely rewritten.

Regarding the results, they are well presented. Consider a more visual output (graphic or figure). It should point out the results more rapidly

Reply – the figures were added.

In the discussion, authors considered the PAPE effect more pronounced during upper limb resistance exercise under BFR only because of the lack of differences in MP and MV in the CONT group. Would not the absolute differences (seems that were not tested for differences) between the conditions be sufficient to point out that more pronounced effect? As in all sets, in all variables, the BFR protocol values are always higher than control.

Reply - due to the change in statistical analysis, much of the discussion has been changed. We introduced statistical differences between particular groups as well as information related with higher power output for the BFR condition compared to the CONT.

At line 201 erase “to” in “during to the rest”

Reply – the change has been made.

At line 205, for further information check Patterson et al. (2019).

Reply – we added comparisons in accordance with recommendations of Patterson et al. (2019).

The limitations are very descriptive and specified, helping on future research. Please pay attention to the state authors make at line 179 “The metabolic stress associated with training under BFR conditions, was not achieved in the current study, due to the short duration of effort in our exercise protocol”, added by the 5 min inter-set rest; and then at line 216 “Therefore, the use of BFR in upper limbs cannot affect the metabolic stress in the pectoralis major and deltoid muscles.” If the short duration and long inter-set rest precluded the metabolic stress of the blood flow restricted muscles, it might do the same to the pectoralis major and deltoid muscles.

Reply – we agree with the reviewer that two statements deny themselves. Changes have been made to make the issue clear and more understandable.

An interesting part of this study that will raise many doubts for further investigations is the fact that BFR did not allow its biggest argument, which is metabolic stress since the restriction was applied for a short time and especially because the intervals between sets were long. This doubt will be kept and should be explored in the future.

Reply – thank you for this comment. We suggest this issue should be considered in future studies. (L 278-280)

Would the authors consider not measuring muscle temperature a limitation?

Reply – we didn’t measure the differences in muscle temperature between particular conditions. We added such information to the limitation of the study.

A section of practical applications will be welcome for readers and practitioners.

Reply – we added practical implications. (L 282-296)

Reviewer 2 Report

I thank the authors for the opportunity to review their work. I reviewed this same paper recently and my major concerns will still not addressed. The idea itself is interesting but the statistical approach is incorrect because there is no comparison between the two groups. A within subject change in one condition has to be compared to the within subject change in the other condition…the difference in differences. It is also stated that the Levene’s test was used, which is not a test that you would run with a within subject design.  

Tour statistical analyses needs to be completed so that the conditions can be compared (Condition x Time Repeated Measures ANOVA). In addition, please state how the effect size was calculated.

Author Response

Thank you for your thoughtful and careful review of the manuscript. We believe that your suggestions and our revision will improve the quality of the manuscript. You will find your comments below followed by our point-by-point responses.

I thank the authors for the opportunity to review their work. I reviewed this same paper recently and my major concerns will still not addressed. The idea itself is interesting but the statistical approach is incorrect because there is no comparison between the two groups. A within subject change in one condition has to be compared to the within subject change in the other condition…the difference in differences. It is also stated that the Levene’s test was used, which is not a test that you would run with a within subject design.

Reply - we didn’t have a chance to explain why we decide to use the one-way ANOVA.

Reply – Thanks for this insightful comment. In submitting the article we used a one-way ANOVA to show the effect of PAP between sets for the BFR condition, without the influence of other aspects, such as the difference between particular conditions. Because the purpose of the study was to determine whether PAP occurs following BFR, not whether the PAP effect increases under BFR compared to Control conditions. Generally the control group was analyzed to show how the PAP effect occurs without BFR. However we agree with the Editor as well as with the reviewer that “randomized crossover design” indicates the need for comparison between groups. Therefore after a broader analysis, we decided to use a 2-way ANOVA (as suggested by one of the reviewers). Additionally according to Wei et al.,( 2012) we also conducted the T-test to show the pure difference between particular sets. In fact, if a main effect occurred for only one factor, multiple comparisons among the variable for this factor are required to identify which specific differences (please, see Wei et al, 2012). Following this indication, we have performed a t-test analysis only between sets under condition;

Wei et al. Comparisons of treatment means when factors do not interact in two-factorial studies. Amino Acids. 2012 May;42(5):2031-5. doi: 10.1007/s00726-011-0924-0.

Your statistical analyses needs to be completed so that the conditions can be compared (Condition x Time Repeated Measures ANOVA). In addition, please state how the effect size was calculated.

Reply – changes have been made. The ref was added

We also made changes based on previously review.

Reviewer 3 Report

The work by Wilk et al. has the purpose to investigate the influence of blood flow restriction (BFR) in post activation performance enhancement (PAPE). PAPE has been recently recognised as a new phenomenon associated to enhanced voluntary force production following high intensity voluntary conditioning activity (CA) (Cuenca-Fernandez et al, Appl Physiol Nutr Metab. 2017). PAPE is clearly separate from other types of post CA muscle potentiation as treppe effect, PTP and PAP; it is characterized by voluntary CA, voluntary muscle function enhancement and late onset (6-7 minutes). Being a new and underknown physiological effect, works and studies investigating PAPE’s features are critically needed and this manuscript adds important evidences to our knowledge about this topic.

Authors designed a crossover randomized study including 10 males. They performed two different excercise protocols, with and without blood flow restriction (BFR vs CONT), in 3 sets of bench press excercise at 70%1RM per each protocol. Study design and methods are well described and seem to be adeguate to the pre-specified purpose. Their work reports that BFR is significantly associated to PAPE compared to control group during set 2 bench excercise. As authors reported, in both CONT and BFR groups increased values of peak power output (PP) and peak bar velocity (PV) occurred, but only the BFR group showed also increased values of mean power output (MP) and mean bar velocity (MV). Reading the work and cited references, it’s not clear the relationship existing between these four variables, their increase referred to starting values and the PAPE. Authors concluded that the significant changes in MP and MV occurring in the BFR group described a more pronounced PAPE effect. It would be interesting to know if, in current literature, these variables used by Wilk et al. have been compared to others to define the most effective ones in detecting PAPE.

In the discussion paragraph there are some major issues that authors should further explain.

They correctly report that this study is underpowered to detect the physiological origins of this phenomenon. Referring to a study not yet published by themselves (reference n° 18 in the manuscript), they refuse the hypothesis that metabolic stress may play a role in PAPE effect during BFR. This cannot be verified due to the inability to check the reported reference.

About the same topic, they wrote: “we can assume that BFR can have a stimulating effect on sensitivity of actin and myosin to Ca2+ ions availability”. There is not a clear relationship between study results and this affirmation. Moreover, the association between Ca2+ sensitivity and PAPE is far to be demonstrated as showed in a recent review by Blazevich and Babault (Front Physiol. 2019).

Further on, authors reported that BFR could mimic effects induced by ischemic preconditioning and supported this conclusion citing a work by Souza et al. (J Strenght Cond Res 2019). This citation may be inappropriate. In their work Souza et al. showed that both high pressure (HP) and low pressure (LP) cuffing were associated to enhanced resistance exercise performance. Since LP cuffing is not able to generate significant arterial occlusion, the authors concluded that ischemic preconditioning couldn’t explain the results and a placebo effect couldn’t be ruled out.

This manuscript adds new data to our knowledge about a recently described phenomenon as PAPE. Reading their work, many further questions rise. In BFR group, PP, PV, MP and MV were all significantly increased in set 2 compared to set 1 and significantly decreased in set 3 compared to set 2. This significant decrease wasn’t reported in CONT group and it would be interesting if authors could comment this data. Is it possible that benefits coming from a more pronounced PAPE in set 2 could be counterbalanced by more pronounced muscle exhaustion in set 3? Could this aspect have significative consequences in real life training programs?

Author Response

Reviewer 3

Thank you for your thoughtful and careful review of the manuscript. We believe that your suggestions and our revision will improve the quality of the manuscript. You will find your comments below followed by our point-by-point responses.

The work by Wilk et al. has the purpose to investigate the influence of blood flow restriction (BFR) in post activation performance enhancement (PAPE). PAPE has been recently recognised as a new phenomenon associated to enhanced voluntary force production following high intensity voluntary conditioning activity (CA) (Cuenca-Fernandez et al, Appl Physiol Nutr Metab. 2017). PAPE is clearly separate from other types of post CA muscle potentiation as treppe effect, PTP and PAP; it is characterized by voluntary CA, voluntary muscle function enhancement and late onset (6-7 minutes). Being a new and underknown physiological effect, works and studies investigating PAPE’s features are critically needed and this manuscript adds important evidences to our knowledge about this topic.

Authors designed a crossover randomized study including 10 males. They performed two different excercise protocols, with and without blood flow restriction (BFR vs CONT), in 3 sets of bench press excercise at 70%1RM per each protocol. Study design and methods are well described and seem to be adeguate to the pre-specified purpose. Their work reports that BFR is significantly associated to PAPE compared to control group during set 2 bench excercise. As authors reported, in both CONT and BFR groups increased values of peak power output (PP) and peak bar velocity (PV) occurred, but only the BFR group showed also increased values of mean power output (MP) and mean bar velocity (MV). Reading the work and cited references, it’s not clear the relationship existing between these four variables, their increase referred to starting values and the PAPE. Authors concluded that the significant changes in MP and MV occurring in the BFR group described a more pronounced PAPE effect. It would be interesting to know if, in current literature, these variables used by Wilk et al. have been compared to others to define the most effective ones in detecting PAPE.

Reply - Commonly, in research on the PAPE effect, the analysis both mean and peak values of power as well velocity are considered.

In the discussion paragraph there are some major issues that authors should further explain.

They correctly report that this study is underpowered to detect the physiological origins of this phenomenon. Referring to a study not yet published by themselves (reference n° 18 in the manuscript), they refuse the hypothesis that metabolic stress may play a role in PAPE effect during BFR. This cannot be verified due to the inability to check the reported reference.

Reply – as it was written in the limitations of study the “there was no analysis of direct physiological changes that would be the basis for explaining the obtained results”. However we observed significant differences between sets during the BFR condition what was also shown in a previous study (ref. 18). The manuscript was (ref 18) recently published doi: 10.1519/JSC.0000000000003649 but this study compared the difference in power output and bar velocity between particular conditions in a single set, without comparison between sets, thus this article is not related with PAPE or PAP effects.

About the same topic, they wrote: “we can assume that BFR can have a stimulating effect on sensitivity of actin and myosin to Ca2+ ions availability”. There is not a clear relationship between study results and this affirmation. Moreover, the association between Ca2+ sensitivity and PAPE is far to be demonstrated as showed in a recent review by Blazevich and Babault (Front Physiol. 2019).

Reply – we agree with the reviewer that this sentence is a too far-reaching speculation. We decided to delete this sentence.

Further on, authors reported that BFR could mimic effects induced by ischemic preconditioning and supported this conclusion citing a work by Souza et al. (J Strenght Cond Res 2019). This citation may be inappropriate. In their work Souza et al. showed that both high pressure (HP) and low pressure (LP) cuffing were associated to enhanced resistance exercise performance. Since LP cuffing is not able to generate significant arterial occlusion, the authors concluded that ischemic preconditioning couldn’t explain the results and a placebo effect couldn’t be ruled out.

Reply – we agree with the reviewer, and we decided to remove this part of the discussion.

This manuscript adds new data to our knowledge about a recently described phenomenon as PAPE. Reading their work, many further questions rise. In BFR group, PP, PV, MP and MV were all significantly increased in set 2 compared to set 1 and significantly decreased in set 3 compared to set 2. This significant decrease wasn’t reported in CONT group and it would be interesting if authors could comment this data. Is it possible that benefits coming from a more pronounced PAPE in set 2 could be counterbalanced by more pronounced muscle exhaustion in set 3? Could this aspect have significant consequences in real life training programs?

Reply – We completely agree with the suggestion regarding the differences between BFR and CONT in the third sets. An analysis of these changes has been added to the discussion. (L 217-230)

Round 2

Reviewer 2 Report

I appreciate the opportunity to review the revised work from the authors. I have a few additional general comments and some additional comments to improve the data presentation.

  • Line 59: you state that BFR increases mechanical tension and elevated metabolic stress. Please be specific about how these are mechanisms..they are seemingly related as it is discussed in the blood flow restriction literature…build up of metabolites, fatigue cross-bridges..requiring additional recruitment of fibers…every fiber that gets activated sufficiently signals to grow/adapt…that’s not really clear from just stating “tension”

https://journals.lww.com/techortho/Abstract/2018/06000/Mechanisms_of_Blood_Flow_Restriction__The_New.2.aspx

  • Please be specific with how you calculated Cohen’s D…with paired data it should be the mean change/ SD of the change

Often times people pool the Pre and Post SD…but that is not the variability of interest

  • You saw interactions with PP and PV…the rest were condition effects…meaning on average BFR produced greater values (from what I can tell).

Therefore…on the ones with no interaction…the analysis should not be interpreted across sets…the main effect of condition would be (BFR vs CON)..with BFR being the average of BFR_Set1, BFR_Set2, BFR_Set3 and same with Control…it will essentially be a paired t-test on the collective average of each. This could also be noted on the figure.

For Table 2…this should only consist of PP and PV…because your statistical test has already informed us that the rest do not change differently across sets.

However, one thing that should be added here…is the comparison between BFR and CON…for example

You currently have

BFR

Set 2 vs Set 1

Set 3 vs Set 1

Set 3 vs Set 2

Con

Set 2 vs Set 1

Set 3 vs Set 1

Set 3 vs Set 2

Given that you  have an interaction…we know that these changes differ some place…but you only interpret within group changes in your results…so you need another 3 rows for the difference between those conditions

BFR – CON (the difference between them for the following sets)

Set 2 vs Set 1

Set 3 vs Set 1

Set 3 vs Set 2

Then you have fully unpacked your interaction. This might change discussion section slightly (probably not much though)

  • In Table 2…it would be useful to include the mean change and variability of that change along with the p values you have.

  • Line 248: this does not appear to be true for muscle…until you get to very low loads…please revise.

https://pubmed.ncbi.nlm.nih.gov/26137897/

  • Line 257: year for Wilk needs to be fixed

  • Line 265: if true, why does pectoralis major grow?

https://pubmed.ncbi.nlm.nih.gov/20618358/

Author Response

Reply – Thank you very much for your comments, I believe that the changes I have made will meet the reviewers expectations. Please find an itemized reply to all your comments, along with the changes we have done on the manuscript.

Line 59: you state that BFR increases mechanical tension and elevated metabolic stress. Please be specific about how these are mechanisms..they are seemingly related as it is discussed in the blood flow restriction literature…build up of metabolites, fatigue cross-bridges..requiring additional recruitment of fibers…every fiber that gets activated sufficiently signals to grow/adapt…that’s not really clear from just stating “tension”

https://journals.lww.com/techortho/Abstract/2018/06000/Mechanisms_of_Blood_Flow_Restriction__The_New.2.aspx

Reply – changes have been made. (L 63-66)

Please be specific with how you calculated Cohen’s D…with paired data it should be the mean change/ SD of the change

Often times people pool the Pre and Post SD…but that is not the variability of interest

Reply - now I understand the details of this question. The SD was independent to each value

You saw interactions with PP and PV…the rest were condition effects…meaning on average BFR produced greater values (from what I can tell).

Therefore…on the ones with no interaction…the analysis should not be interpreted across sets…the main effect of condition would be (BFR vs CON)..with BFR being the average of BFR_Set1, BFR_Set2, BFR_Set3 and same with Control…it will essentially be a paired t-test on the collective average of each. This could also be noted on the figure.

Reply – The information about the main condition difference was added. (L172-174)

Also was made t-test for differences between condition for set 1, set 2, set 3. (Table 4)

For Table 2…this should only consist of PP and PV…because your statistical test has already informed us that the rest do not change differently across sets.

Reply - for full transparency of results, we decided to leave the full results.

However, one thing that should be added here…is the comparison between BFR and CON…for example

You currently have

BFR

Set 2 vs Set 1

Set 3 vs Set 1

Set 3 vs Set 2

Con

Set 2 vs Set 1

Set 3 vs Set 1

Set 3 vs Set 2

Given that you have an interaction…we know that these changes differ some place…but you only interpret within group changes in your results…so you need another 3 rows for the difference between those conditions

BFR – CON (the difference between them for the following sets)

Set 2 vs Set 1

Set 3 vs Set 1

Set 3 vs Set 2

Then you have fully unpacked your interaction. This might change discussion section slightly (probably not much though)

Reply - statistical analysis and results have been changed as suggested, as well as discussion.

In Table 2…it would be useful to include the mean change and variability of that change along with the p values you have.

Reply – done (table 3)

Line 248: this does not appear to be true for muscle…until you get to very low loads…please revise.

https://pubmed.ncbi.nlm.nih.gov/26137897/

Reply – I agree with the reviewer, we delete this sentence, as well as w added new sentence. (L270 – 271)

Line 257: year for Wilk needs to be fixed

Reply – change has been made

Line 265: if true, why does pectoralis major grow?

https://pubmed.ncbi.nlm.nih.gov/20618358/

Reply - The study indicated by the reviewer concerns long-term changes, while the presented study analyzes acute changes. However, we added a sentence regarding the reviewer's suggestion.

(L – 287 -290)